# Highly redundant neuropeptide volume co-transmission underlying episodic activation of the GnRH neuron dendron

Xinhuai Liu[1], Shel-Hwa Yeo[1,2], H James McQuillan[1], Michel K Herde[1], Sabine Hessler[1], Isaiah Cheong[1], Robert Porteous[1], Allan E Herbison[1,2]*

[1]Centre for Neuroendocrinology and Department of Physiology, University of Otago School of Biomedical Sciences, Dunedin, New Zealand; [2]Department of Physiology, Development and Neuroscience, University of Cambridge, Cambridge, United Kingdom

**Abstract** The necessity and functional significance of neurotransmitter co-transmission remains unclear. The glutamatergic 'KNDy' neurons co-express kisspeptin, neurokinin B (NKB), and dynorphin and exhibit a highly stereotyped synchronized behavior that reads out to the gonadotropin-releasing hormone (GnRH) neuron dendrons to drive episodic hormone secretion. Using expansion microscopy, we show that KNDy neurons make abundant close, non-synaptic appositions with the GnRH neuron dendron. Electrophysiology and confocal GCaMP6 imaging demonstrated that, despite all three neuropeptides being released from KNDy terminals, only kisspeptin was able to activate the GnRH neuron dendron. Mice with a selective deletion of kisspeptin from KNDy neurons failed to exhibit pulsatile hormone secretion but maintained synchronized episodic KNDy neuron behavior that is thought to depend on recurrent NKB and dynorphin transmission. This indicates that KNDy neurons drive episodic hormone secretion through highly redundant neuropeptide co-transmission orchestrated by differential post-synaptic neuropeptide receptor expression at the GnRH neuron dendron and KNDy neuron.

*For correspondence:
aeh36@cam.ac.uk

Competing interests: The authors declare that no competing interests exist.

## Introduction

Many neurons use the co-transmission of classical small-molecule and neuropeptide neurotransmitters to signal within their networks. Such co-transmission enables a wide dynamic range of signaling through the frequency coding of transmitter release (*van den Pol, 2012*; *Vaaga et al., 2014*; *Tritsch et al., 2016*). However, the extent and functional significance of co-transmission remains unclear in most forebrain circuits. For example, 'Dale's Principle', as formulated by *Eccles, 1976*, posits that all axons of an individual neuron will release the same set of transmitters. The generality of this concept has now been challenged for small-molecule co-transmitters (see *Tritsch et al., 2016*) but remains untested for neuropeptide co-transmission.

The kisspeptin neurons located in the arcuate/infundibular nucleus of the mammalian hypothalamus appear to engage in substantial co-transmission being glutamatergic and synthesizing at least four neuropeptides including kisspeptin, NKB, dynorphin, and galanin (*Lehman et al., 2010*; *Skrapits et al., 2015*). In a range of mammals, the majority of these arcuate nucleus (ARN) neurons co-express kisspeptin, NKB, and dynorphin resulting in their 'KNDy' moniker (*Lehman et al., 2010*; *Skrapits et al., 2015*). Immunohistochemical studies have demonstrated that the three neuropeptides are packaged within separate vesicles within KNDy nerve terminals (*Lehman et al., 2010*; *Murakawa et al., 2016*). Although the KNDy neurons project widely throughout the limbic system (*Krajewski et al., 2010*; *Yeo and Herbison, 2011*; *Yip et al., 2015*), they are best characterized as being the 'GnRH pulse generator' responsible for episodically activating the gonadotropin-releasing

hormone (GnRH) neurons to drive pulsatile luteinizing hormone (LH) secretion (*Clarkson et al., 2017*; *Herbison, 2018*; *Plant, 2019*). The KNDy neurons are proposed to achieve this by providing an episodic stimulatory input to the distal projections of the GnRH neurons close to their secretory zone in the median eminence (ME) (*Herbison, 2018*). These distal projections of the GnRH neuron have shared features of dendrites and axons and have been termed 'dendrons' (*Herde et al., 2013*; *Herbison, 2016*; *Moore et al., 2018b*; *Yip et al., 2021*). The nature and functional significance of KNDy co-transmission at the GnRH neuron dendron remains unknown.

## Results

### KNDy neurons form abundant close appositions with GnRH neuron distal dendrons

We first established the anatomical relationship between KNDy fibers and the GnRH neuron distal dendrons in the ventrolateral ARN using confocal immunohistochemistry. Analysis of para-horizontal sections revealed numerous kisspeptin-expressing fibers passing through and around the GnRH neuron dendrons as they turned toward the ME (*Figure 1A*). We found that 68.5 ± 6.2% of analyzed GnRH dendrons segments had at least one apposition with a kisspeptin fiber (N = 4 female mice). To examine kisspeptin fibers originating from KNDy neurons, we assessed the relationship of GnRH neuron dendrons to fibers co-expressing kisspeptin and NKB. In total, we observed 4.0 ± 0.4 kisspeptin close appositions/100 μm of dendron length and 67.0 ± 7.6% of these co-expressed NKB (*Figure 1A*; N = 4). Prior work has shown that the GnRH neuron projections are innervated by KNDy neurons in addition to preoptic area kisspeptin neurons that do not express NKB (*Yip et al., 2015*). On average, each kisspeptin/NKB fiber made close appositions with 4.5 ± 0.7 GnRH neuron dendrons. This arrangement suggests an abundant, divergent innervation of the GnRH neuron distal dendrons by KNDy neurons.

### KNDy neurons signal through volume transmission to the GnRH neuron dendrons

While regular confocal analysis is useful for assessing anatomical relationships, it is unable to unambiguously define synapses. Expansion microscopy (ExM) uses isotropic swelling of the tissue specimen to provide ~70 nm spatial resolution that can reliably image synapses in the brain (*Wassie et al., 2019*). We have previously demonstrated using ExM that a 'side-plane' overlap of >0.23 μm (0.95 μm post-expansion) or face-plane (z-stack) overlap >0.42 μm (1.75 μm post-expansion) between a synaptophysin-immunoreactive bouton and the GFP within the cytoplasm of a GnRH neuron represents a bona fide synapse (*Wang et al., 2020*). The presence of kisspeptin synapses on distal dendrons was examined by assessing synaptophysin-containing kisspeptin boutons opposed to GFP-expressing dendrons. Surprisingly, we identified no kisspeptin-containing synaptic profiles on 45 individual distal dendrons (>15 μm length each, N = 3 mice) with all kisspeptin/synaptophysin-immunoreactive boutons being outside the criteria for a synapse or indeed quite separate from the GFP-expressing dendron (*Figure 1B,C*). The average distance between kisspeptin–synaptophysin boutons and GnRH neuron dendrons was 2.22 ± 0.27 μm (post-expansion). Nevertheless, many synaptophysin-expressing boutons without kisspeptin were identified to make synaptic appositions with GnRH neuron dendrons (density of 2.3 ± 0.1 synaptophysin synapses per 10 μm dendron) (*Figure 1B*). This suggested that kisspeptin inputs to the distal dendron did not make conventional synapses. To verify this, we used the same approach to examine the morphological relationship of kisspeptin inputs to the GnRH neuron soma/proximal dendrites in the rostral preoptic area, where local kisspeptin neurons form conventional synapses with GnRH neurons (*Piet et al., 2018*). We assessed eight 60 μm lengths of GFP soma/dendrite in each of three mice and found many synaptophysin–kisspeptin boutons forming synapses with GFP-expressing dendrites (*Figure 1D*). Overall, 37.9 ± 0.8% of all synaptophysin boutons synapsing on GnRH neuron soma/proximal dendrites (0.81 ± 0.44 per 10 μm) contained kisspeptin.

These observations indicate that while kisspeptin inputs to the GnRH cell bodies and proximal dendrites exist as classical synapses, this is not the case for the distal dendron where divergent KNDy signaling occurs through short-diffusion volume transmission.

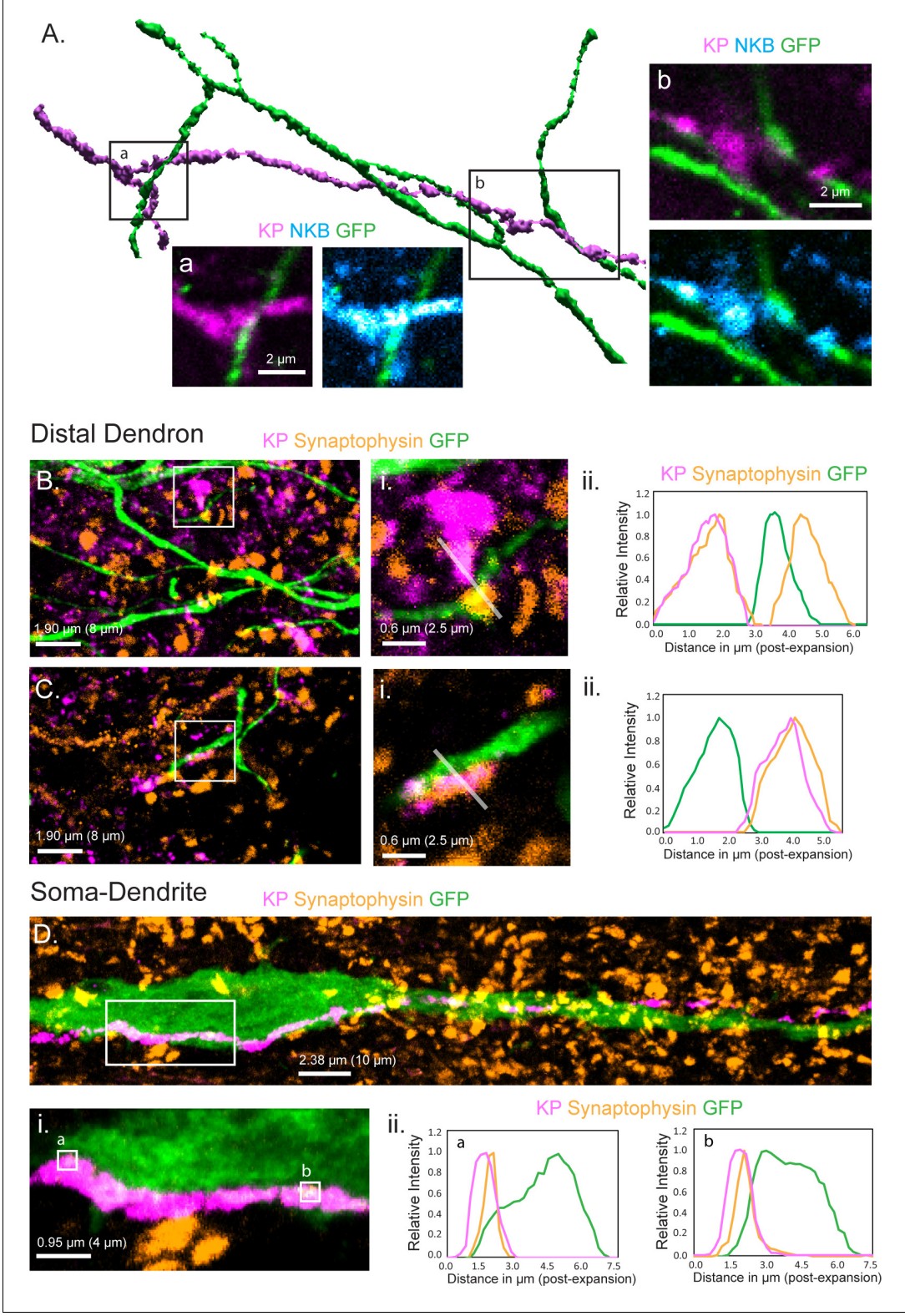

**Figure 1.** Relationship of KNDy neuron fibers to GnRH neuron distal dendrons. (**A**) 3D reconstruction of regular confocal images showing a KNDy fiber expressing kisspeptin (KP) and neurokinin B (NKB) making close appositions with three GnRH neuron dendrons in the ventrolateral ARN of a GnRH-GFP mouse. (**B,C**) Expansion microscopy views of GnRH distal dendrons surrounded by kisspeptin fibers and synaptophysin puncta. Insets (i) highlight two examples of synaptophysin-expressing kisspeptin terminals adjacent to GnRH dendrons. Gray lines

*Figure 1 continued on next page*

*Figure 1 continued*

indicate the line scans used to generate the fluorescence relative intensity profiles shown to the right. (Bii) A GFP-expressing dendron with a chemically unidentified synapse on one side (>0.95 μm overlap between synaptophysin and GFP signals) and a kisspeptin terminal making a close non-synaptic contact on the other (no overlap). (Cii) Another example of kisspeptin terminal (kisspeptin and synaptophysin) making a close non-synaptic (overlap <0.95 μm) contact with a GnRH dendron. (D) Expansion microscopy view of a GnRH neuron cell body and proximal dendrite surrounded by synaptophysin puncta and with a kisspeptin fiber running along its length. Imaging in the z-axis face view shows two locations (ia and b) where kisspeptin/synaptophysin puncta make synapses on the GnRH neuron cell body. (Dii) Fluorescence relative intensity profiles show two synaptophysin-containing kisspeptin boutons exhibiting >1.75 μm overlap with cytoplasmic GFP of the GnRH neuron. Scale bars show pre-expansion units with post-expansion values in brackets.

## GnRH neuron dendrons only respond to one of the four KNDy co-transmitters

We have previously established an acute horizontal brain slice preparation (*Figure 2A*) in which changes in [Ca$^{2+}$] within the thin GnRH neuron distal dendrons can be measured providing a proxy for electrical activity (*Iremonger et al., 2017*). A thick horizontal brain slice containing the ME and adjacent hypothalamic tissue was prepared from adult male and diestrous-stage female *Gnrh1-Cre* mice previously given preoptic area injections of AAV9-CAG-FLEX-GCaMP6s (*Figure 2B*). To mimic the episodic release of transmitters, multi-barreled pipettes were used to apply candidate neuro-transmitters as short 90 s puffs to the region of the GnRH neuron dendrons (*Figure 2B*), while recording calcium signals from multiple dendrons simultaneously using confocal imaging. In vivo recordings show that KNDy neurons exhibit synchronized episodes of activity for 1–2 min prior to each LH pulse (*Han et al., 2019*; *McQuillan et al., 2019*). In controls, where we applied artificial cerebrospinal fluid (aCSF) at different distances and pressures from the brain slice, we found that slight movement artifacts were unavoidable. Placing a pipette 30–130 μm above the surface of the brain slice and puffing aCSF resulted in a 4.5 ± 0.5% (mean ± SD) increase in [Ca$^{2+}$] within the dendrons beneath (*Figure 3A,B*). As such, we defined a threshold for a drug-induced change in [Ca$^{2+}$]

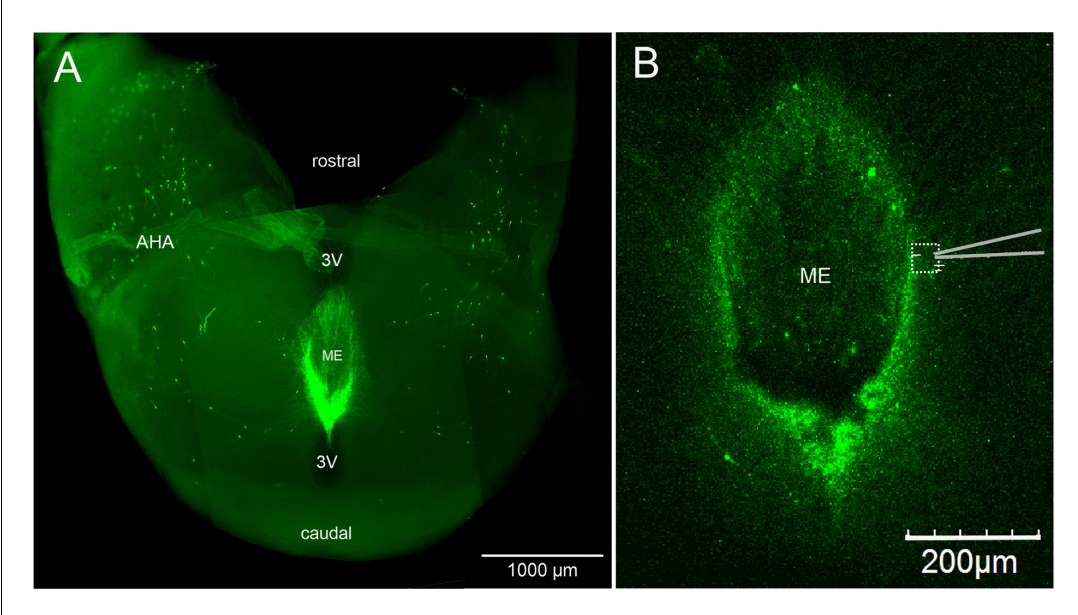

**Figure 2.** Horizontal brain slice preparation for examining GnRH neuron distal dendrons. (A) View looking down on a thick horizontal brain slice prepared from a *Gnrh*-GFP mouse showing the laterally positioned GnRH neuron cell bodies in the anterior hypothalamic area (AHA) and the concentrated GnRH neuron projections in the median eminence (ME). 3V, third ventricle. (B) Higher-power view of the same orientation of the ME region in a living brain slice prepared from a GCaMP6s AAV-injected *Gnrh1-Cre* mouse showing the recording location (dotted square) and position of the puff pipette.

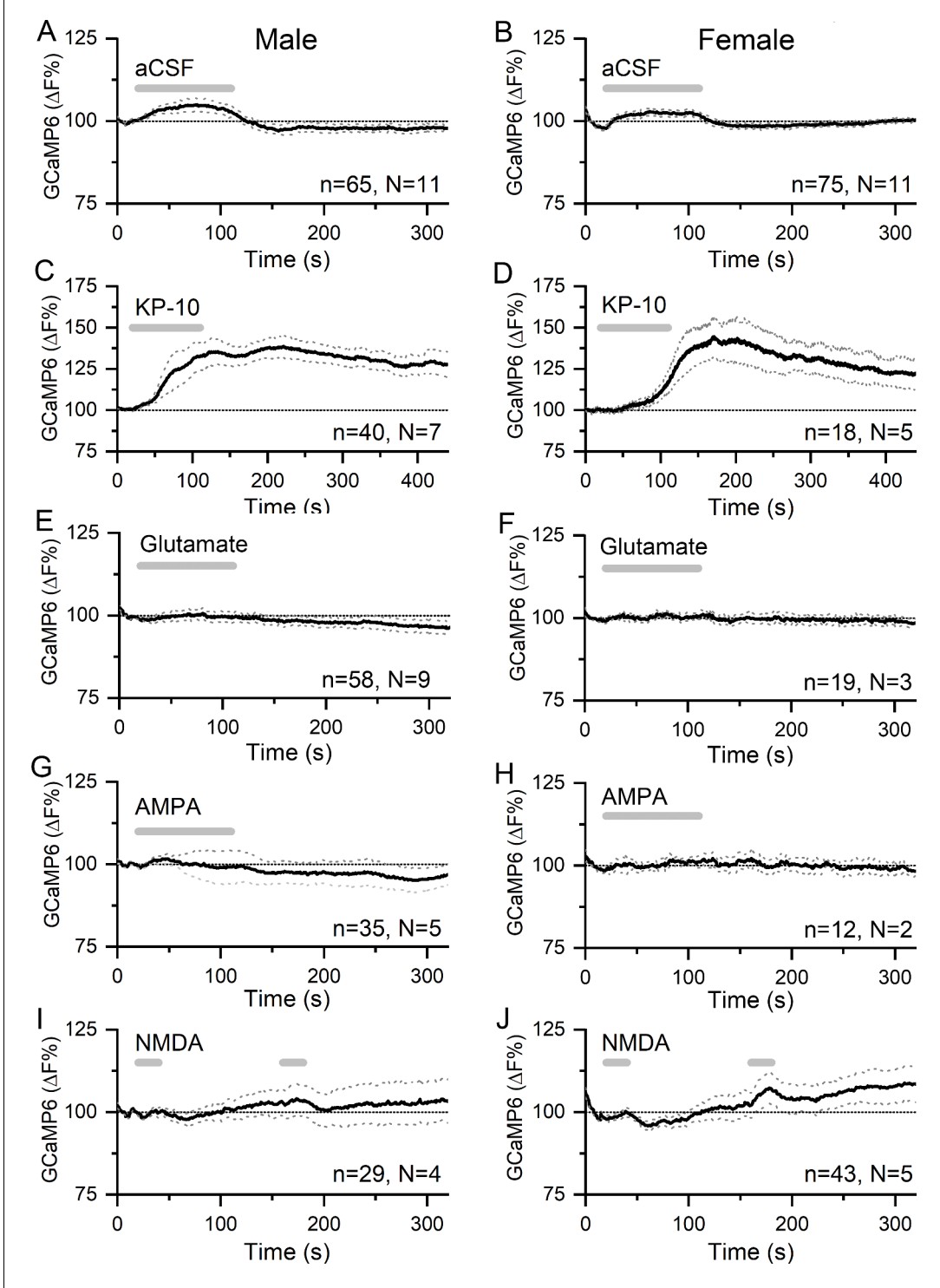

**Figure 3.** Kisspeptin but not glutamate regulates [Ca$^{2+}$] in GnRH neuron distal dendrons. (**A,B**) Effects of 90 s puffs of aCSF on GCaMP6 fluorescence in GnRH neuron distal dendrons in male and female *Gnrh1-Cre::GCaMP6s* mice. (**C,D**) Puffs of kisspeptin-10 (100 nM) generate large, sustained increases in [Ca$^{2+}$] in both sexes. Note the altered x- and y-axes. (**E–J**) Long (90 s) or short (20 s) puffs of glutamate (600 µM), AMPA (80 µM), and NMDA (200 µM) have no significant effects on [Ca$^{2+}$] in dendrons. Dotted lines indicate 95% confidence intervals. Numbers of dendrons (n) and mice (N) are given for each treatment and each sex.

as requiring an increase or decrease greater than the aCSF mean plus two standard deviations from the control change (i.e. >5.5%) and a response that outlasted the time of the puff.

Application of a 90 s puff of kisspeptin (100 nM) generated a 35.1 ± 0.1% (mean ± SEM, male, median 35.1%) to 40.5 ± 0.2% (female, median 40.1%) rise in dendron [Ca²⁺] that peaked and then gradually subsided across the 400 s duration of the recording in both male (N = 7) and female (N = 5) mice (*Figure 3C,D*). Approximately 91% of dendrons responded to kisspeptin.

In contrast, 90 s puffs of glutamate (600 µM) that would activate both ionotropic and metabotropic receptors were found to have no significant effects on dendron [Ca²⁺] in either males (N = 9) or females (N = 2) (*Figure 3E,F*). Furthermore, 90 s puffs of AMPA (80 µM) had no effects on dendron [Ca²⁺] in males (N = 5) or females (N = 2) (*Figure 3G,H*). In further experiments (N = 2 males, N = 2–3 females), glutamate and AMPA were given as two shorter 30 s puffs but were also found to have no effects on dendron fluorescence (not shown). Similarly, N-methyl-D-aspartate (NMDA )(200 µM) given as two 30 s puffs in the absence of Mg²⁺ had no effects in males (N = 4) or females (N = 5) (*Figure 3I–J*).

We next tested the effects of the co-expressed KNDy neuropeptides NKB and dynorphin on the GnRH dendron. Ninety second puffs of 100 nM NKB generated small rises in dendron [Ca²⁺] (male, 2.45 ± 0.04% [mean ± SEM], median 2.54%; female, 4.73 ± 0.05%, median 4.64%) that were not significantly different to control aCSF puffs in either males (N = 11) or females (N = 5) (*Figure 4A,B*). To ensure that this was not a technical false negative, horizontal brain slices were prepared from *Kiss1^Cre/+*;GCaMP6f mice in the same manner. Identical puffs of 100 nM NKB directly above KNDy neurons were found to exert potent stimulatory effects on [Ca²⁺] in both males (N = 3) and females (N = 3) (*Figure 4C,D*). The effects of NKB on KNDy neuron [Ca²⁺] were more potent in males than in females (area under curve = 18,542 ± 2101 s. %F versus 5441 ± 1158, p<0.001, Mann–Whitney test) (*Figure 4C,D*).

Puffs of 200 nM dynorphin (90 s) were also found to have no significant effect on dendron [Ca²⁺] in either males (N = 11) or females (N = 11) (male, 2.83 ± 0.03%, median 2.86%; female, 2.14 ± 0.05%, median 2.19%) (*Figure 5A,B*). As an inhibitory neuromodulator, it was possible that any suppressive actions of dynorphin may be difficult to assess on basal dendron [Ca²⁺]. As such, we tested

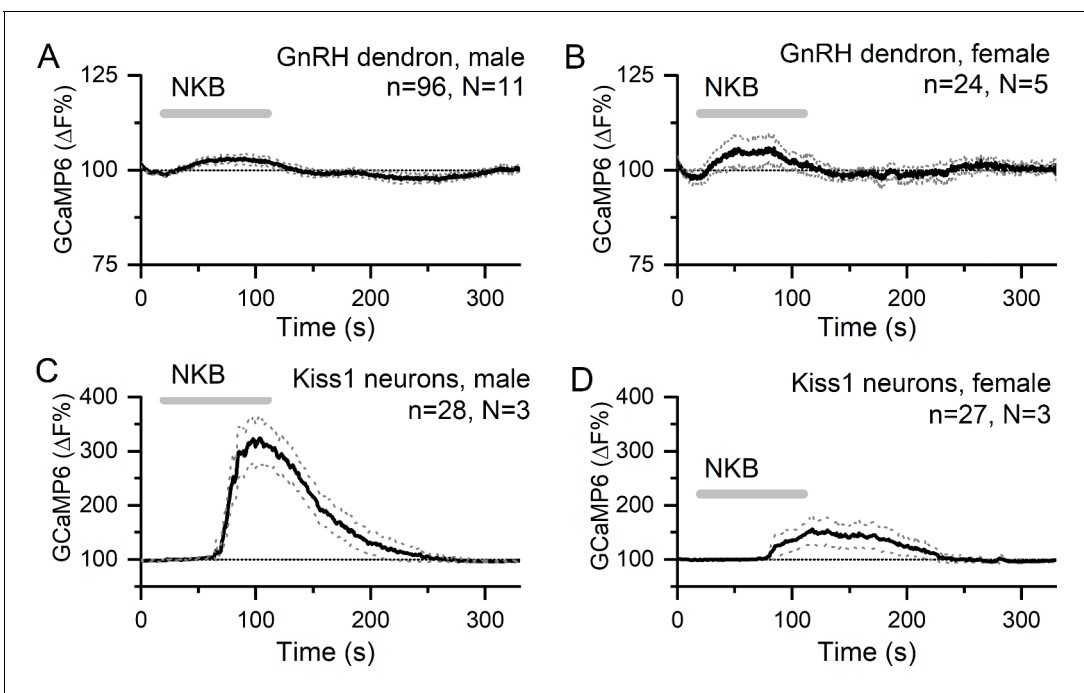

**Figure 4.** NKB increases [Ca²⁺] in KNDy neurons but not in GnRH neuron distal dendrons. (A,B) Ninety second puffs of 100 nM NKB have no significant effect on GCaMP6 fluorescence in GnRH neuron distal dendrons in male and female *Gnrh1-Cre::GCaMP6s* mice. (C,D) Ninety second puffs of 100 nM NKB evoke large increases in [Ca²⁺] in KNDy neurons of male and female *Kiss1^Cre/+*;*GCaMP6f* mice. Note the altered y-axis. Dotted lines indicate 95% confidence intervals. Numbers of dendrons (n) and mice (N) are given for each treatment and each sex.

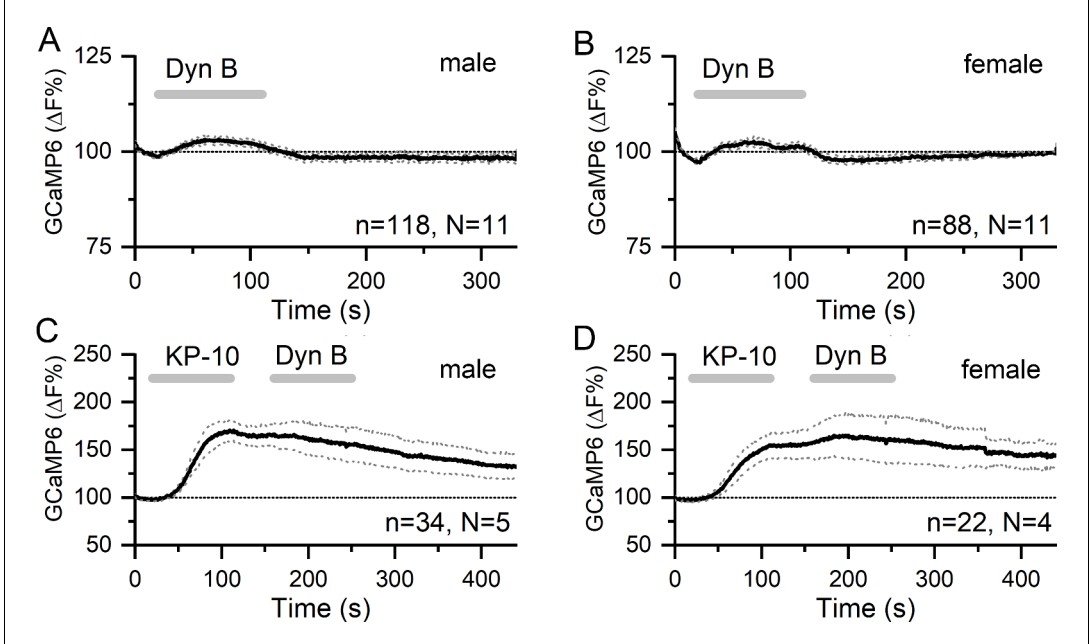

**Figure 5.** Dynorphin has no effect on [Ca²⁺] in GnRH neuron distal dendrons. (A,B) Ninety second puffs of 200 nM dynorphin have no significant effect on GCaMP6 fluorescence in GnRH neuron distal dendrons in male and female *Gnrh1-Cre::GCaMP6s* mice. (C,D) Similarly, dynorphin has no effect on kisspeptin-10-evoked increases in GCaMP6 fluorescence in either sex. Dotted lines indicate 95% confidence intervals. Numbers of dendrons (n) and mice (N) are given for each treatment and each sex.

the effect of dynorphin on kisspeptin-evoked increases in dendron [Ca²⁺]. However, dynorphin continued to fail to alter dendron [Ca²⁺] in males (N = 5) and females (N = 4) (*Figure 5C,D*).

## NKB and dynorphin are released from KNDy terminals in the vicinity of the GnRH dendrons

Our immunohistochemical studies indicate a high level of co-expression between kisspeptin and NKB at KNDY boutons apposing GnRH neuron dendrons (*Figure 1A*). To examine whether NKB and dynorphin are actually released from these terminals, we tested whether ARN neurons in the immediate vicinity of the dendrons might respond to NKB and dynorphin released from adjacent KNDy terminals. AAV9-DIO-hChR2-mCherry was injected into the ARN of *Kiss1^{Cre/+}*;*Rosa26*-tdT mice to transduce KNDy neurons with channelrhodopsin. We then prepared acute brain slices from these mice and made cell-attached recordings from unidentified, non-kisspeptin neurons (no red fluorescence) in the region of the dendrons and examined the effects of optogenetic activation of KNDy neurons with 10 and 20 Hz blue light. Qui and colleagues had previously shown that these were the optimal ChR2 stimulation frequencies to evoke NKB and dynorphin release from KNDy terminals in the ARN (*Qiu et al., 2016*).

Initially, KNDy neurons from three mice (two diestrus and one male) were patched and the effects of 10 and 20 Hz blue light stimulation determined. A near-perfect fidelity between blue light activation and action currents was found for both 10 and 20 Hz stimulations (n = 8, in three mice; *Figure 6A*). Next, in these same mice, 40 unidentified ventrolateral ARN neurons surrounded by KNDy fibers were patched and the effects of 10 and 20 Hz stimulation examined. In total, 11 cells responded to blue light activation with nine exhibiting a slow-onset and prolonged (1–6 min, median 3 min) excitation (*Figure 6B*) and another two cells showing a similar temporal profile of suppressed firing (*Figure 6C*). The ChR2-evoked increases in firing rate were reversibly inhibited by exposure to the NK3R antagonist SB222200 in five of nine cells (*Figure 6B*) and inhibitions were reversibly blocked by the kappa-opioid antagonist nor-binaltorphimine (nor-BNI ) (*Figure 6C*). In two diestrous mice in which ChR2 transduction failed (no response of eight KNDy cells to blue light), no effects of blue light were detected on the firing of 21 unidentified ventrolateral ARN neurons (not shown).

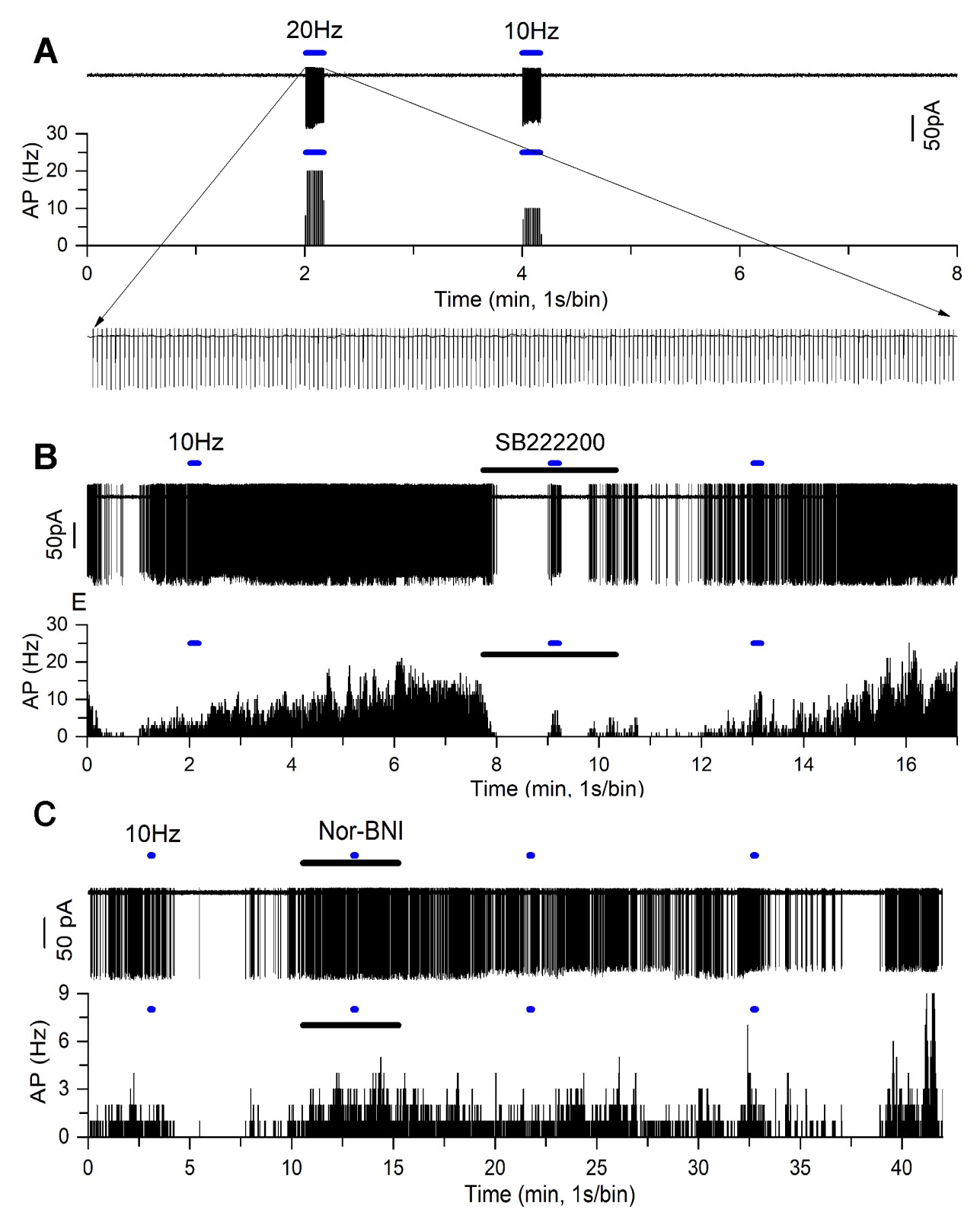

**Figure 6.** NKB and dynorphin are released from KNDy neuron terminals in the vicinity of the GnRH neuron dendrons. (**A**) Optogenetic blue light (473 nm) activation of a transduced KNDy neuron at 20 Hz and 10 Hz. Rate meter trace below shows sustained 20 Hz and 10 Hz firing during the majority of the 10 s stimuli. Expanded inset shows individual action currents. (**B**) Cell-attached recording (above) and rate meter histogram (below) of an unidentified ventrolateral ARN neuron exhibiting a slow excitatory response to 10 Hz blue light (blue bar) that is reversibly suppressed by addition of

*Figure 6 continued on next page*

*Figure 6 continued*

the NK3R antagonist SB222200 (20 μM) shown as the dark bar. (**C**) Cell-attached recording of another unidentified ventrolateral ARN neuron exhibiting an inhibitory response to 10 Hz blue light (blue bar) that is reversibly suppressed by addition of the kappa-opioid antagonist nor-binaltorphimine (NBI) (12.5 μM) shown as the dark bar. The inhibitory response to optogenetic activation returns after ~20 min.

## Kisspeptin signaling in KNDy neurons is not necessary for KNDy neuron synchronization

The experiments reported above indicate that, of the co-transmitters released from KNDy terminals, the GnRH neuron dendrons only express functional receptors for kisspeptin. This predicts that kisspeptin is the only co-transmitter released by KNDy terminals necessary to activate the GnRH neuron dendron and drive pulsatile LH secretion. This is in striking contrast to signaling at the KNDy cell body where the opposite relationship exists with kisspeptin having no effect on excitability (*de Croft et al., 2013*), while the co-released neuropeptides NKB and dynorphin directly excite and inhibit firing, respectively (*de Croft et al., 2013*; *Ruka et al., 2013*; *Qiu et al., 2016*). It is proposed that the recurrent NKB and dynorphin signaling at the KNDy cell body underlies their synchronized episodic behavior (*Qiu et al., 2016*; *Moore et al., 2018a*). Thus, it appears that only kisspeptin signaling is required at the dendron, while kisspeptin is redundant at the recurrent collaterals. To test these hypotheses in vivo, we generated mice in which only kisspeptin was deleted from KNDy neurons and assessed both KNDy neuron synchronization (transmission at the KNDy cell body) and pulsatile LH secretion (transmission at the KNDy terminals).

In the homozygous state, the $Kiss1^{Cre/Cre}$ mice used in this study represent a *Kiss-1* deletion (*Yeo et al., 2016*). To characterize these mice further, immunohistochemical analyses of adult female null $Kiss1^{Cre/Cre}$ mice (N = 4) confirmed the complete absence of kisspeptin peptide in cells or fibers within the ARN while immunoreactivity for NKB remained (*Figure 7A–C*).

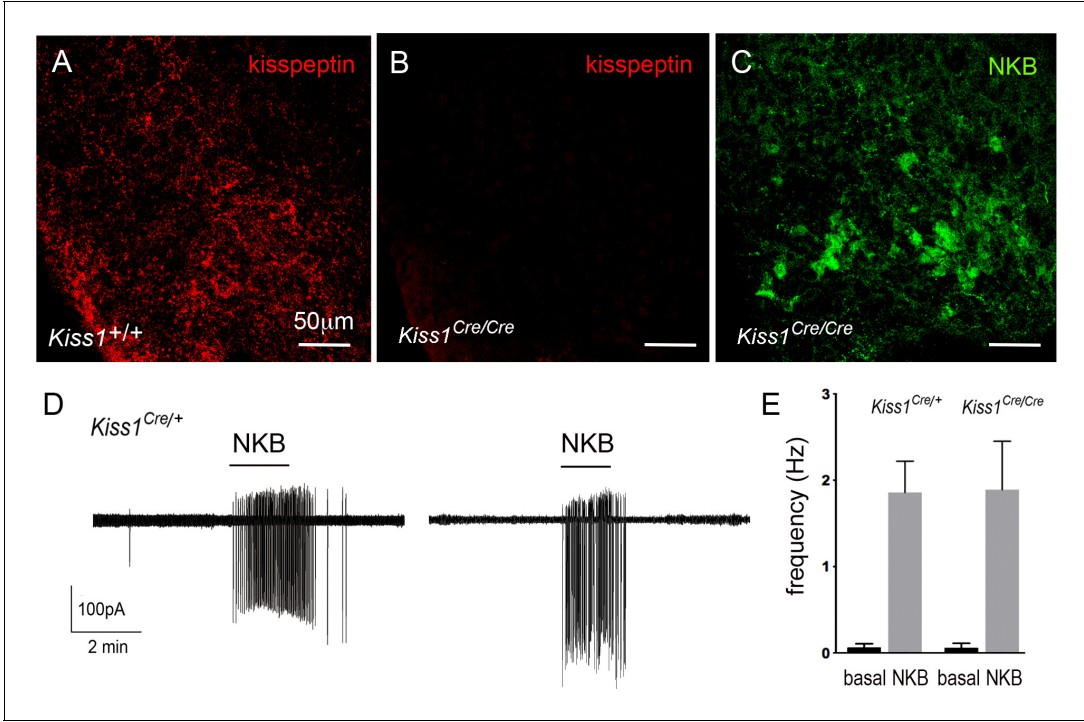

**Figure 7.** Characterization of *Kiss1*-null mice. (**A-C**) Immunofluorescence for kisspeptin (**A,B**) and NKB (**C**) in the ARN of wild-type (**A**) and *Kiss1*-null (**B, C**) female mice. (**D**) Cell-attached recordings showing the effects of 100 nM NKB on firing of KNDy neurons in acute brain slices prepared from female heterozygous $Kiss1^{Cre/+}$ and homozygous (null) $Kiss1^{Cre/Cre}$;*Rosa26*-tdT mice. (**E**) Mean ± SEM changes in KNDy neuron firing evoked by 100 nM NKB in heterozygous $Kiss1^{Cre/+}$ and homozygous (null) $Kiss1^{Cre/Cre}$;*Rosa26*-tdT mice.

We also assessed NKB receptor function at the KNDy neuron cell body in these mice by undertaking cell-attached recordings of KNDy neurons in the acute brain slice and comparing the effects of 50 nM NKB on KNDy neuron firing in control heterozygous $Kiss1^{Cre/+}$;$Rosa26$-tdT mice (N = 4) and null $Kiss1^{Cre/Cre}$;$Rosa26$-tdT (N = 5) mice. Both lines showed the same very low spontaneous firing rates (0.06 ± 0.04 Hz, n = 14, $Kiss1^{Cre/+}$; 0.06 ± 0.06 Hz, n = 14, $Kiss1^{Cre/Cre}$) typical of KNDy neurons (*de Croft et al., 2012*), and 50 nM NKB exerted the same marked stimulatory effect on firing (1.9 ± 0.4 Hz, n = 14, $Kiss1^{Cre/+}$, 1.9 ± 0.6 Hz, n = 14, $Kiss1^{Cre/Cre}$) (*Figure 7D,E*). Together, these studies indicate that the expression of NKB and function of NKB receptors in KNDy neurons is normal in the absence of kisspeptin in $Kiss1$-null mice.

The synchronized episodic activity of the ARN kisspeptin neurons can be measured in real time using in vivo GCaMP6 fiber photometry (*Han et al., 2018*). Twenty-four hour GCaMP fiber photometry recordings from the middle/caudal ARN of adult female $Kiss1^{Cre/Cre}$;$Rosa26$-tdT mice (N = 4) given prior injections of AAV9-CAG-FLEX-GCaMP6s, revealed the presence of frequent abrupt ARN kisspeptin neuron synchronizations (*Figure 8A*). This pattern is similar to that observed in ovariectomized heterozygous $Kiss1^{Cre/+}$;$Rosa26$-tdT mice (*Figure 8C*) and compatible with homozygous $Kiss1$-$^{Cre/Cre}$ mice being hypogonadal (*Yeo et al., 2016*).

## Kisspeptin in KNDy neurons is essential for episodic activation of the GnRH neuron dendron to generate pulsatile LH secretion

The in vitro studies above indicate that kisspeptin is the only co-transmitter signaling from KNDy neurons to the GnRH neuron dendrons. We reasoned that if this was the case, then the synchronized output of KNDy neurons, which is normally perfectly correlated with pulsatile LH secretion (*Han et al., 2019*; *McQuillan et al., 2019*), would fail to signal to the GnRH neuron dendron in $Kiss1$-null mice and result in the absence of pulsatile LH secretion. AAV-injected $Kiss1^{Cre/Cre}$ mice (N = 4) underwent fiber photometry recordings while also having 5 min tail-tip bleedings performed for 1 hr to assess pulsatile LH secretion. As the nearest possible control, ovariectomized heterozygous AAV-injected $Kiss1^{Cre/+}$ mice (N = 3) were assessed at the same time. Despite robust ARN$^{KISS}$ neuron synchronization events (SEs), all four $Kiss1$-null mice exhibited an invariant, very low level of LH demonstrating a complete uncoupling of the KNDy pulse generator from LH secretion (*Figure 8B*). In contrast, all three ovariectomized $Kiss1^{Cre/+}$ mice exhibited pulsatile LH secretion that was perfectly correlated with ARN kisspeptin neuron SEs (*Figure 8C*).

## Discussion

Co-transmission in the brain is considered to result primarily from the differential release of synaptic vesicles containing separate neurotransmitters (*van den Pol, 2012*; *Vaaga et al., 2014*; *Tritsch et al., 2016*). The KNDy neuron provides an example of co-transmission in which its output at recurrent collaterals and its primary efferent target are differentially interpreted by converse patterns of post-synaptic neuropeptide receptor expression (*Figure 9*). Kisspeptin is the only co-transmitter active at the GnRH neuron dendron, whereas the exact opposite situation exists at KNDy neuron recurrent collaterals where all of the co-transmitters, except kisspeptin, are active (*Navarro et al., 2011*; *de Croft et al., 2013*; *Ruka et al., 2013*; *Qiu et al., 2016*). Thus, KNDy neurons conform to Dale's Principle of uniform transmitter expression across their axonal arbor but solve the problem of differential signaling through opposite patterns of post-synaptic receptor expression at their targets (*Figure 9*). While this type of receptor-dependent co-transmission has been reported for small-molecule transmitters, it has not been shown for co-released neuropeptides (*Tritsch et al., 2016*). We demonstrate that this mode of co-transmission is physiologically relevant in vivo as deletion of kisspeptin from the repertoire of transmitters used by KNDy neurons abolishes episodic hormone secretion while maintaining KNDy neuron synchronization behavior.

As a result of their migration from the nose into the brain at mid-gestation, the GnRH neuron cell bodies are scattered throughout the basal forebrain (*Wray, 2010*). The functional difficulties imposed by this topography appear to be solved, in part, by GnRH neurons focusing their blended dendritic/axonal projections on the ME where they can be regulated in a concerted manner by the KNDy pulse generator (*Herbison, 2018*). While it is clear that classic synaptic inputs exist on the GnRH neuron dendron (*Moore et al., 2018b*; *Wang et al., 2020*), we note that this is not the case for the KNDy neuron innervation. Although many close appositions were identified with regular

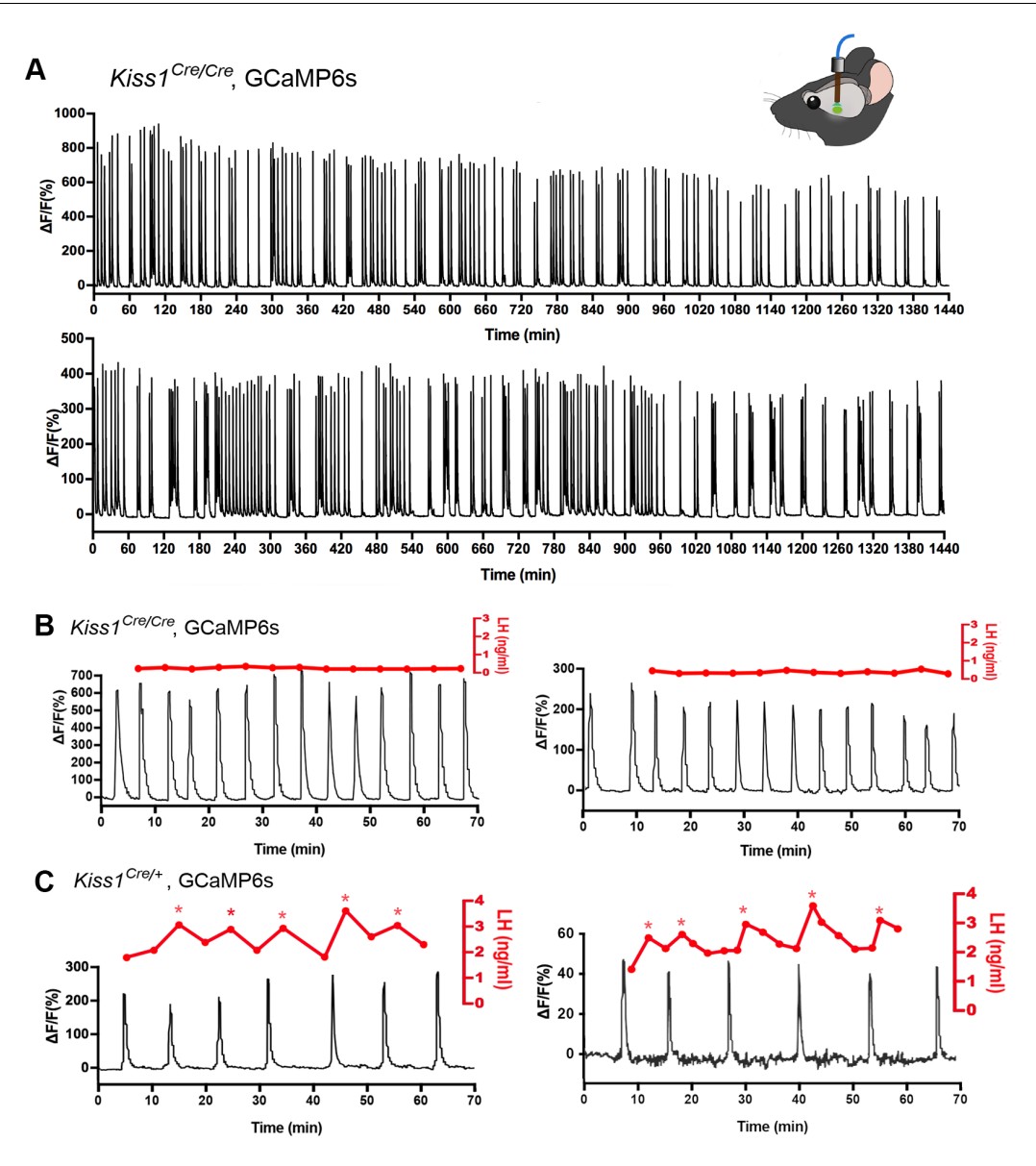

**Figure 8.** Mice with deleted Kiss1 exhibit KNDY neuron synchronization events but fail to generate pulsatile LH secretion. (**A**) Representative examples of 24 hr in vivo GCaMP6 fiber photometry recordings of KNDy neuron synchronization events from two female *Kiss1^{Cre/Cre}*; *Rosa26*-tdT::*GCaMP6s* mice. (**B**) Representative examples of combined 5 min tail-tip bleeding for LH levels (red) and GCaMP6 fiber photometry (black) recordings from two female *Kiss1^{Cre/Cre}*; *Rosa26*-tdT::*GCaMP6s* mice. (**C**) Representative GCaMP6 photometry, 3–5 min tail-tip bleeding LH levels from two ovariectomized heterozygous female *Kiss1^{Cre/+}*;*Rosa26*-tdT::*GCaMP6s* mice.

confocal analysis, no evidence was found for these to be synaptic inputs between KNDy fibers and GnRH neuron dendrons using ExM. This was despite the ExM identification of bona fide kisspeptin synapses at the GnRH neuron cell body and proximal dendrites. An electron microscopic investigation in the rat also reported that closely apposed kisspeptin fibers and GnRH neuron nerve terminals in the ME do not form synapses (*Uenoyama et al., 2011*). This morphological relationship is indicative of short-distance volume transmission (*van den Pol, 2012*). We also note that individual KNDy fibers form appositions with multiple GnRH neuron dendrons. Thus, it seems probable that the synchronous activation of GnRH neuron dendrons is achieved by volume transmission originating from multiple boutons of KNDy neuron axons winding their way through GnRH neuron dendrons.

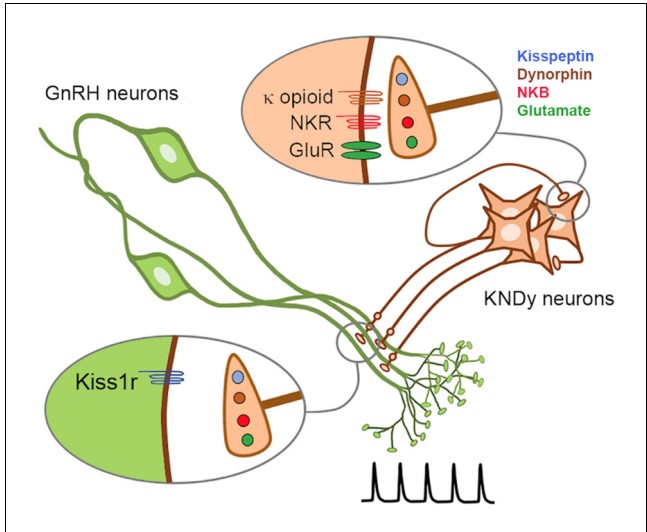

**Figure 9.** Schematic diagram depicting the proposed patterns of co-transmission that occur at the KNDy neuron recurrent collaterals (upper inset) and at their non-synaptic projections to the GnRH neuron dendrons where only kisspeptin is active (lower inset). NKR, neurokinin receptors; GluR, glutamate receptors.

The nature of KNDy signaling at the GnRH neuron has been difficult to discern. Studies in the mouse show that KNDy neurons only project to the GnRH neuron distal projections in an around the ME (*Yip et al., 2015*; *Qiu et al., 2016*). Although immunohistochemical studies have indicated that GnRH neurons express receptors for NKB and/or dynorphin (*Krajewski et al., 2005*; *Weems et al., 2016*), this has not been supported by mRNA profiling and electrophysiological studies (*Mitchell et al., 1997*; *Sannella and Petersen, 1997*; *Navarro et al., 2011*; *Qiu et al., 2016*). We find here that NKB had no effects on [Ca$^{2+}$] in GnRH neuron dendrons, indicating that they are not likely to express NK3R. This is in contrast with the KNDy neurons themselves where NKB evokes a strong activation in the same brain slice preparation. We also found no evidence for dynorphin to modulate either basal or kisspeptin-evoked [Ca$^{2+}$] in the GnRH dendron. The latter result does not support the hypothesis that dynorphin signaling is involved in terminating the kisspeptin-evoked GnRH pulse at the level of the GnRH neuron, at least in mice (*Weems et al., 2018*). Notably, a very recent study has shown that nitric oxide can terminate and re-set kisspeptin responses at the GnRH neuron cell body (*Constantin et al., 2021*), and it is possible that a similar scenario exists at the level of the GnRH neuron dendron.

Surprisingly, we observed that glutamate, AMPA, and NMDA had no impact on [Ca$^{2+}$] in GnRH neuron dendrons. Our initial investigations found that puffs of glutamate to the dendron evoked small but significant responses (*Herde et al., 2013*). However, in those early studies we were limited technically to examining only the dendrons of the few GnRH neurons that have cell bodies close to the ME. We also noted that glutamate responses were markedly less effective as we puffed along the projection with puffs greater than 350 microns from the cell body being ineffective (*Herde et al., 2013*). With the present technical approach, we are able to examine the distal dendrons of the vast majority GnRH neurons that are located in the preoptic area and have their cell bodies located 1000s of microns from the ME. Hence, it seems very likely that there is falling gradient of glutamate receptor expression along the GnRH neuron projections such that, while glutamate will activate those rare dendrons coming from GnRH neuron somata close by the ME, 99% of distal dendrons have no glutamate receptors. Functionally, it is notable that low-frequency KNDy neuron activation, that would be presumed to release only glutamate, has no impact on LH secretion (*Han et al., 2015*).

As KNDy neurons package their individual neurotransmitters into separate vesicles (*Murakawa et al., 2016*), they could differentially traffic specific vesicles to distinct axonal projections. However, this does not appear to occur at their two key efferent outputs for regulating pulsatile LH secretion as KNDy neuron recurrent collaterals and their projections to GnRH neuron

dendrons all co-express kisspeptin, NKB, and dynorphin (*Lehman et al., 2010*; *Figure 1*). In agreement, we find evidence that kisspeptin, NKB, and dynorphin are all released from KNDy terminals adjacent to the GnRH neuron dendron, while Qui and colleagues demonstrated that NKB and dynorphin were released at recurrent collaterals innervating the KNDy cell bodies (*Qiu et al., 2016*). Together, these studies indicate that KNDy neurons conform to 'Dale's Principle' (*Eccles, 1976*) and that differential signaling is achieved at these two key target sites by selective post-synaptic neuropeptide receptor expression.

These observations also suggest that a population of unidentified neurons located in the ventrolateral ARN may have their activity entrained by the GnRH neuron pulse generator. We found evidence for both NKB- and dynorphin-receptive neurons in this area responding to KNDy neuron activation as well as cells excited in a NK3R-independent manner that may be activated by kisspeptin (*Fu and van den Pol, 2010*; *Liu and Herbison, 2015*). As yet, there have been no endocrine or behavioral outputs identified that match the abrupt temporal dynamics of pulsatile LH secretion and it will be intriguing to identify the nature of these cells in due course.

In summary, evidence indicates that the GnRH pulse generator network uses converse patterns of post-synaptic receptor expression at the two main targets of the KNDy neuron required for episodic hormone secretion. This likely solves the problem of signal resolution generated when using neuropeptide volume transmission in the close vicinity of the KNDy somata and GnRH neuron dendrons. Although highly redundant at the level of the post-synaptic space, this novel pattern of co-transmission enables neurons to achieve differential signaling at varied targets without the need for selective trafficking of neuropeptide transmitters throughout their axonal arbor.

# Materials and methods

## Key resources table

| Reagent type (species) or resource | Designation | Source or reference | Identifiers | Additional information |
|---|---|---|---|---|
| Genetic reagent (*M. musculus*) | STOCK Tg(Gnrh1-cre) 1Dlc/J | Jackson Laboratory | Stock #: 021207 RRID:IMSR_JAX:021207 | |
| Genetic reagent (*M. musculus*) | B6.DBA-Tg(Gnrh1-EGFP)1Phs | Spergel et al., doi:10.1523/JNEUROSCI.19-06-02037.1999 | MGI:6158458 | |
| Genetic reagent (*M. musculus*) | Kiss1$^{tm2(CreGFP)Coll}$: tdTom | Yeo et al., doi:org/10.1111/jne.12435 2016 | | |
| Genetic reagent (*M. musculus*) | B6.Cg-Gt(ROSA)26 Sor$^{tm9(CAG-tdTomato)Hze}$/J (Ai9) | Jackson Laboratory | Stock #: 07909 RRID:IMSR_JAX:007909 | |
| Genetic reagent (*M. musculus*) | B6J.Cg-Gt(ROSA)26 Sor$^{tm95.1(CAG-GCaMP6f)Hze}$/MwarJ (Ai95D) | Jackson Laboratory | Stock #: 028865 RRID:IMSR_JAX:028865 | |
| Transfected construct (*M. musculus*) | AAV9-CAG-FLEX-GCaMP6s-WPRE-SV40 | Penn Vector Core | RRID:Addgene_100844 | $1.7 \times 10^{-13}$ GC/mL |
| Transfected construct (*M. musculus*) | AAV9-EF1-DIO-hChR2-(H134R)-mCherry-WPRE-hGH | Penn Vector Core | RRID:Addgene_20297 | $4.4 \times 10^{13}$ GC/mL |
| Antibody | Anti-Kisspeptin 10 (polyclonal rabbit) | Alain Caraty, INRA, France | Cat#: AC566 RRID:AB_2314709 | (1:2000) |
| Antibody | Anti-Kisspeptin 10 (polyclonal sheep) | Alain Caraty, INRA, France | Cat#: AC024 | (1:8000) |
| Antibody | Anti-NKB (polyclonal rabbit) | Novus Biologicals | Cat#: NB300-201 RRID:AB_10000783 | (1:5000) |
| Antibody | Anti-NKB (polyclonal guinea pig) | P. Ciofi; INSERM; France | Cat#: IS-3/63 RRID:AB_2732894 | (1:5000) |

*Continued on next page*

*Continued*

| Reagent type (species) or resource | Designation | Source or reference | Identifiers | Additional information |
|---|---|---|---|---|
| Antibody | Anti-GFP (polyclonal chicken) | Abcam | Cat#: AB13970 RRID:AB_300798 | (1:8000) |
| Antibody | Anti-Synaptophysin 1 (polyclonal guinea pig) | Synaptic Systems | Cat#: 101004 RRID:AB_1210382 | (1:800) |
| Antibody | Goat anti-chicken (polyclonal goat, Alexa488-conjugate) | ThermoFisher Scientific | Cat#: A-11039 RRID:AB_2534096 | (1:400) |
| Antibody | Goat anti-guinea pig (polyclonal goat, biotin-conjugated) | Vector Laboratories | Cat#: BA-7000 RRID:AB_2336132 | (1:400) |
| Antibody | Goat anti-rabbit (polyclonal goat, ATTO647N-conjugated) | Sigma–Aldrich | Cat#: 40839 RRID:AB_1137669 | (1:400) |
| Antibody | AffiniPure F(ab')$_2$ Fragment Donkey Anti-Sheep (polyclonal donkey, biotin-conjugated) | Jackson ImmunoResearch Labs | Cat#: 713-066-147 RRID:AB_2340717 | (1:1500) |
| Antibody | Alexa Fluor 488 F(ab')$_2$ Fragment Donkey Anti-Rabbit (H+L) | Jackson ImmunoResearch Labs | Cat#: 711-546-152 RRID:AB_2340619 | (1:1000) |
| Antibody | Alexa Fluor 488 Goat anti-chicken IgY (H+L) | ThermoFisher | Cat#: A-11039 RRID:AB_2534096 | (1:200) |
| Antibody | Alexa Fluor 568 goat anti-guinea pig IgG (H+L) | Invitrogen | Cat#: A-11075 RRID:AB_2534119 | (1:200) |
| Antibody | Alexa Fluor 633 goat anti-rabbit IgG (H+L) | Invitrogen | Cat#: A-21071 RRID:AB_2535732 | (1:200) |
| Chemical compound, drug | (RS)-α-amino-3-hydroxy-5-methyl-4-isoxazolepropionic acid (AMPA) | Tocris | Cat#: 0169 | 80 µM |
| Chemical compound, drug | D-2-Amino-5-phosphonopentanoic acid sodium salt (D-AP5) | Tocris | Cat#: 0106/1 | 50 µM |
| Chemical compound, drug | DL-2-Amino-5-phosphonopentanoic acid sodium salt (DL-AP5) | Tocris | Cat#: 3693/10 | 25 µM |
| Chemical compound, drug | 6-Cyano-7-nitroquinoxaline-2,3dione disodium (CNQX) | Tocris | Cat#: 1045/1 | 10 µM |
| Chemical compound, drug | Dynorphin B (Dyn B) | Tocris | Cat#: 3196/1 | 100–200 nM |
| Chemical compound, drug | SR 95531 hydrobromide (GABAzine) | Tocris | Cat#: 1262/10 | 5 µM |
| Chemical compound, drug | Glutamate acid | Sigma–Aldrich | Cat#: G1251 | 300 or 600 µM |
| Chemical compound, drug | Kisspeptin-10 (KP-10) | Calbiochem | Cat#: 45888 | 100 nM |
| Chemical compound, drug | Neurokinin B (NKB) | Tocris | Cat#: 1582/1 | 50–100 nM |
| Chemical compound, drug | N-Methyl-D-aspartic acid (NMDA) | Tocris | Cat#: 0114 | 100 or 100 µM |
| Chemical compound, drug | Tetrodotoxin (TTX) | Alomone Labs | Cat#: T-550 | 0.5–2 µM |
| Chemical compound, drug | SB 222200 | Tocris | Cat#: 1393/10 | 20 µM |

*Continued*

| Reagent type (species) or resource | Designation | Source or reference | Identifiers | Additional information |
|---|---|---|---|---|
| Chemical compound, drug | Nor-binaltorphimine | Tocris | Cat#: 0347/10 | 12.5 µM |
| Software, algorithm | ImageJ image analysis software | ImageJ (https://imagej.net/) | RRID:SCR_003070 | |
| Software, algorithm | Vaa3D data visualization software | Vaa3D (http://www.vaa3d.org) | RRID:SCR_002609 | |

## Animals

C57BL/6 *Gnrh-GFP* mice (*Spergel et al., 1999*), C57BL/6J *Gnrh1-Cre* mice (JAX stock #021207) (*Yoon et al., 2005*), 129S6Sv/Ev C57BL/6 *Kiss1$^{Cre/+}$* mice (*Yeo et al., 2016*) alone or crossed on to the Ai9 *Rosa26*-CAG-LSL-tdTomato$^{+/-}$ reporter line (JAX stock #07909) (*Madisen et al., 2010*) (*Kiss1$^{Cre/+}$;Rosa26*-tdT mice) or Ai95 (RCL-GCaMP6f)-D line (JAX stock #028865) (*Madisen et al., 2015*) (*Kiss1$^{Cre/+}$;GCaMP6f* mice) were group-housed in individually-ventilated cages with environmental enrichment under conditions of controlled temperature (22 ± 2°C) and lighting (12 hr light/12 hr dark cycle; lights on at 6:00 hr and lights off at 18:00 hr) with ad libitum access to food (Teklad Global 18% Protein Rodent Diet 2918, Envigo, Huntingdon, UK) and water. Daily vaginal cytology was used to monitor the estrous cycle stage. All animal experimental protocols were approved by the University of Otago, New Zealand (96/2017) or the University of Cambridge, UK (P174441DE).

## Stereotaxic surgery

Adult mice (2–4 months old) were anaesthetized with 2% isoflurane and placed in a stereotaxic apparatus with prior local Lidocaine (4 mg/kg bodyweight, s.c.) and Carprofen analgesia (5 mg/kg body weight, s.c.). A custom-made bilateral Hamilton syringe apparatus holding two syringes with needles held 0.9 mm apart was used to perform bilateral injections into the preoptic area (AP 0.10 mm, depth 4.3 mm) or ARN (AP −2.00 mm, depth 5.9 mm) of *Gnrh1-Cre* or *Kiss1$^{Cre/+}$;Rosa26*-tdT mice, respectively. The needles were lowered into place over 2 min and left in situ for 3 min before the injection was made. For GnRH dendron calcium imaging, 1.5 µL of AAV9-CAG-FLEX-GCaMP6s-WPRE-SV40 (1.7 × 10$^{-13}$ GC/mL, University of Pennsylvania Vector Core) was injected bilaterally into the preoptic area at a rate of ~100 nL/min with the needles left in situ for a further 10 min before being withdrawn. For the optogenetic activation of KNDy neurons, 1 µL of AAV9-EF1-DIO-hChR2-(H134R)-mCherry-WPRE-hGH (4.35 × 10$^{13}$ GC/mL; Penn Vector Core) was injected into the ARN of *Kiss1$^{Cre/+}$; Rosa26*-tdT mice. For GCaMP photometry studies, 1 µL of AAV9-CAG-FLEX-GCaMP6s-WPRE-SV40 (1.7 × 10$^{13}$ GC/mL, University of Pennsylvania Vector Core) was injected into the ARN of *Kiss1$^{Cre/+}$ or Kiss1$^{Cre/Cre}$; Rosa26*-tdT mice that were then implanted with a unilateral indwelling optical fiber (400 µm diameter; 0.48 NA, Doric Lenses, Quebec, Canada) positioned directly above the mid-caudal ARN using the same coordinates for the AAV injections. Carprofen (5 mg/kg body weight, s.c.) was administered for post-operative pain relief. Mice were housed individually for the remaining experimental period. A typical post-surgical period of 4–6 weeks was allowed, with daily handling and habituation to the recording appartus.

## Immunohistochemistry for expansion microscopy

Diestrous-stage *Gnrh-GFP* mice aged between 2 and 4 months old were perfused transcardially with 4% paraformaldehyde in 0.1 M phosphate-buffered saline (PBS). Fifty-micron coronal brain sections were cut on a vibratome and kept in cryoprotectant until used. Sections were pre-treated with 0.1% sodium borohydrate (Sigma–Aldrich) in Tris-buffered saline (TBS) for 15 min at room temperature and then further treated with 0.1% Triton-X-100 (Sigma–Aldrich) and 2% goat serum in TBS overnight at 4°C for improving antibody penetration. Next, sections were washed and incubated for 72 hr at 4°C with a cocktail of chicken anti-GFP (1:8000; Abcam), guinea pig anti-synaptophysin1 (1:800; Synaptic Systems), and rabbit anti-kisspeptin-10 (1:2000; gift from Alain Caraty) antisera added to the incubation solution made up of TBS, 0.3% Triton-X-100, 0.25% bovine serum albumin, and 2% goat serum. All subsequent incubations were performed using the same incubation solution.

Sections were rinsed in TBS, followed by incubation with biotinylated goat anti-guinea pig immunoglobulin (Vector Laboratories) mixed with Alexa488-conjugated goat anti-chicken (ThermoFisher Scientific) and ATTO 647N goat anti-rabbit immunoglobulins (Sigma–Aldrich), 1:400 each, for 15 hr at 4°C. Sections were then expanded as previously reported (*Wang et al., 2020* eLife). Briefly, sections were stained with 1:1000 Hoechst 33342 dye (Thermo Fisher Scientific), processed for linking with anchoring agent, and trimmed to include the rostral preoptic area or ARN region. Next, sections were incubated in monomer solution, followed by gelling solution in a humidified chamber at 37°C. Gel-embedded sections were digested overnight with proteinase K. Following that, the gels were rinsed in PBS, incubated with 1:1500 Streptavidin-568 at 37°C for 3 hr, and rinsed again with PBS. Lastly, water was added every 20 min, up to five times during the expansion step. Expanded samples were placed in imaging chamber filled with water and cover slipped using #1.5 cover glass.

Imaging was undertaken using a Nikon A1R upright confocal microscope equipped with a water-immersing lens (25× numerical aperture 1.1; working distance 2 mm). All images were captured using sequential scanning mode using 500–550 nm, 580–620 nm, and 620–660 nm bandpass filters for Alexa Fluor 488, Alexa Fluor 568, and ATTO 647, respectively. All image stacks (frames: 102.51 × 51.25 μm, 1024 × 512 pixels) were acquired at 0.6 μm focus intervals. Images were analyzed using ImageJ to determine the number of kisspeptin terminals containing synaptophysin opposed to GnRH proximal or distal dendrites. For the cell body/proximal dendrite, 250 μm (60 μm pre-expansion) of contiguous primary dendrite arising from the GnRH cell body was randomly selected from three rostral preoptic area sections in each of the three mice. Each apposing synaptophysin–kisspeptin bouton (diameter > 0.4 μm) was examined to establish the side-on view or a z-stack/face-view of the imaged synapse. A line scan was then performed across this plane, and the relative intensity of the Alexa488 and ATTO647 was measured and plotted in Microsoft Excel. For the face-view orientation, a 2 μm × 2 μm (width × length) box was drawn on each of the synaptophysin–kisspeptin boutons to measure the relative intensity of Alexa488 and ATTO647 through the z-stack. An apposition was considered a synapse where the signals overlapped by >0.95 μm (0.23 μm pre-expansion) in the side-on plane or >1.75 μm (0.42 μm pre-expansion) in the z-stack through/face-view (*Wang et al., 2020*). The same method was used to establish the relationship between synaptophysin–kisspeptin profiles and the distal dendron using 60 μm (15 μm pre-expansion) contiguous lengths of dendron located in the ventrolateral ARN.

## Immunohistochemistry for multi-label fluorescence

Adult diestrous $Kiss1^{Cre/+}$ and $Kiss1^{Cre/Cre}$ mice (2–3 months old) underwent transcardial perfusion. Coronal brain sections of 40 μm thickness were prepared and incubated in rabbit anti-NKB (1:5000; Novus Biologicals) and sheep anti-Kisspeptin 10 antisera (1:8000; gift of Alain Caraty) followed by biotinylated donkey anti-sheep immunoglobulins (1:1500; Jackson Immunoresearch), donkey anti-rabbit conjugated with Alexa Fluor 488 (1:1000, Thermo Fisher Scientific), and Streptavidin Alexa Fluor 647 (1:1500, ThermoFisher Scientific). Images were acquired using a Leica SP8 Laser Scanning Confocal Microscope and a 63× oil immersion objective (numerical aperture 1.20; working distance 300 μm) with image stacks collected at 0.6 μm intervals (Cambridge Advanced Imaging Centre).

Adult *Gnrh-GFP* mice were perfused transcardially and a ventral para-horizontal brain slice prepared by making a single horizontal cut from the base of the brain (*Figure 2*). Brain slices were incubated in rabbit anti-kisspeptin 10 (1:2000; gift of Dr. Alain Caraty), guinea pig anti-NKB (1:5000; IS-3/61, gift of Dr. Philippe Ciofi), and chicken anti-GFP (1:2000; AB16901, Chemicon) antisera followed by goat anti-chicken Alexa Fluor 488 (1:200, A-11039, Thermo Fisher), goat anti-guinea pig Alexa Fluor 568 (1:200, A-11075, Invitrogen), and goat anti-rabbit Alexa Fluor 633 (1:200, A-21071, Invitrogen) immunoglobulins. Sections were examined on a Zeiss LSM 710 confocal microscope with a 63×/1.4 Plan Apochromat objective at 1.4× zoom and Nyquist resolution using ZEN software (version 5.5.0.375). Image stacks were acquired at 0.38 μm intervals. The ventrolateral ARN was imaged at 0.38 μm z-step intervals with four image stacks tiled 2 × 2 taken at 1056 × 1056 pixels resolution and stitched using the 3D Stitching plugin in FIJI in linear blend mode (*Preibisch et al., 2009*). For each animal, 10 regions of >100 μm length of GFP-labeled dendrons selected due to their proximity to the ME and direction of projection toward ME were analyzed. A close apposition was defined as the absence of dark pixels between elements or even a slight overlap of the different channels. For 3D reconstruction, raw data image stacks were isosurface rendered in Amira (version 5.3, Visage Imaging, San Diego, CA) using the neuronal reconstruction plugin by *Schmitt et al., 2004*.

## Confocal imaging of GnRH neuron dendron Ca$^{2+}$ concentration

The procedure for making GCaMP6 recordings from GnRH neuron distal dendrons has been reported previously (*Iremonger et al., 2017*). In brief, the mouse was killed by cervical dislocation, the brain quickly removed, and optic tract peeled off. The dorsal surface of the brain was then glued to a vibratome cutting stage (VT1200s, Leica) and submerged in ice-cold (<2°C) sucrose-containing cutting solution (in mM) (75 NaCl, 75 sucrose, 2.5 KCl, 20 HEPES, 15 NaHCO$_3$, 0.25 CaCl$_2$, 6 MgCl$_2$, 25 D-glucose, bubbled with 95% O$_2$/5% CO$_2$; 320 mOsmol). A single 500 µm thick horizontal slice containing the ME and surrounding tissue (*Figure 2*) was made and incubated in the cutting solution (34 ± 1°C) and then recording aCSF (in mM)(118 NaCl, 3 KCl, 10 HEPES, 25 NaHCO$_3$, 2.5 CaCl$_2$, 1.2 MgCl$_2$, 11 D-glucose; 95% O$_2$/5% CO$_2$; 27 ± 1°C) for at least 1 hr before being transferred to the recording chamber (27 ± 1°C) where the slice was held between two meshes with a perfusion flow rate of 1.5 mL/min. Imaging was performed with an Olympus FV1000 confocal microscope fitted with a 40×, 0.8 NA objective lens and 3× zoom with the aperture fully open. GCaMP was excited with a 488 nm Argon laser and emitted light passed through a 505–605 nm bandpass filter.

Test compounds were dissolved in aCSF and locally puff-applied with a patch pipette (4–6 MΩ) at low pressure (~1 psi) controlled by a Pneumatic Picopump (PV821, World Precision Instruments, USA) for 20–90 s. Total recording time for each test was 500 s. The tip of the puff pipette was positioned 30–130 µm above the surface of the slice. Initially, GnRH neuron dendrons were tested with 30 s puffs of 20 mM KCl to assess their viability and only those fibers displaying fast Ca$^{2+}$ responses were used for experiments. Puffs of aCSF alone could generate small changes in signal with a slow rise and decay that followed the timing of the puff (4.50 ± 0.05%, range of −12% to 12% with a median of 4.44%). Acceptable recordings had to have <5% drift in baseline fluorescence intensity across the time of the experiment. Control aCSF recordings were undertaken throughout the series of experiments, and kisspeptin tests were undertaken as the last challenge. To block action potential-dependent synaptic transmission, TTX (0.5–1 µM; Alamone Labs, Israel) was added to the recording aCSF for the duration of the experiment. For neuropeptide tests, GABAzine (5 µM; Tocris Bioscience, UK), D-AP5 (50 µM, Tocris) or DL-AP5 (25 µM), and CNQX (10 µM, Tocris) were added to the TTX-containing aCSF to eliminate ionotropic GABA$_A$ receptor and glutamate receptor activation. A 0 mM Mg$^{2+}$ aCSF containing TTX was applied at least 5 min before NMDA (200 µM; Tocris) puffs. Glutamate (600 µM; Tocris), AMPA (80 µM; Tocris), neurokinin B (100 nM; Tocris), dynorphin B (100–200 nM; Tocris), and Kisspeptin (100 nM, Calbiochem, USA) were maintained as frozen stock solutions and dissolved into the recording aCSF.

Image acquisition was performed with Fluoview 1000 software. Frame scans (512 × 512 pixels) were performed on zoomed regions at 0.9 Hz frame rate with the lowest possible laser power. Image analysis was performed with Fluoview1000 software and ImageJ. Regions of interest (ROI) were drawn around individual GnRH neuron dendrons. The percentage of GCaMP6 fluorescence was calculated as GCaMP6 ΔF% = 100x(F/F0), where F is the average fluorescence of ROI in each consecutive frame and F0 is the average fluorescence of ROI in first 10–20 frames before test drug puffs. Any tissue drift in the x–y axis was corrected by enlarging the ROI or by using Turboreg (ImageJ). All data are presented as mean± SEM. The average calcium traces are shown with lower and upper 95% confidence levels and have a minimum animal number of 4. Statistical analyses were performed with non-parametric paired sample Wilcoxon signed-rank test or repeated-measures tests (Kruskal–Wallis or Friedman test with a post hoc Dunn's test).

## Acute brain slice electrophysiology

Brain slice electrophysiology was undertaken as reported previously (*Hessler et al., 2020*). In brief, adult male and diestrous-stage female *Kiss1$^{Cre/+}$;Rosa26*-tdT mice were killed by cervical dislocation and 250 µm thick coronal brain slices containing the ARN prepared on a vibratome (VT1000S; Leica) in an ice-cold 75 mM sucrose aCSF cutting solution. Slices were then incubated for at least 1 hr in recording aCSF (in mM) (120 NaCl, 3 KCl, 26 NaHCO$_3$, 1 NaH$_2$PO$_4$, 2.5 CaCl$_2$, 1.2 MgCl$_2$, 10 HEPES, 11 glucose; 95% O$_2$/5% CO$_2$, 32°C) before being transferred to a recording chamber. Loose-seal cell-attached recordings (10–30 MΩ) were made from tomato-expressing KNDy neurons visualized through an upright BX51 Olympus microscope. Cells were visualized by brief fluorescence illumination and approached using infrared differential interference contrast optics. Loose seals were made using recording electrodes (3.5–5.2 MΩ) filled with aCSF and action currents recorded in the voltage

clamp mode with a 0 mV voltage command. Signals were recorded using a Multiclamp 700B amplifier (Molecular Devices, Sunnyvale, CA) connected to a Digidata 1440A digitizer (Molecular Devices) and low-pass filtered at 3 kHz before being digitized at a rate of 10 kHz. For analysis, spikes were detected using the threshold crossing method. Signal acquisition and analysis was carried out with pClamp 10.7 (Molecular Devices). The effects of NKB on KNDy neurons were assessed by adding 50 nM NKB to the aCSF for 1 min.

For brain slice optogenetic studies, blue light (473 nm) was delivered using a Grass S88X Stimulator controlled DPSS laser (Ike-Cool, USA) coupled with a 100 μm diameter fiber optic probe that was placed immediately above the surface of slice. Laser intensity at the tip of the glass pipette surrounding the optic fiber was 4 mW. The blue light pulse was 5 ms wide, and a train of pulses at 10 Hz and/or 20 Hz was given for 10 s to each patched cell. The neuropeptide antagonists SB 222200 (20 μM; Tocris) and nor-binaltorphimine (12.5 μM; Tocris) were added to the recording aCSF.

### In vivo GCaMP photometry and LH pulse bleeding

The GCaMP6 fiber photometry procedure used to detect KNDy neuron SEs has been described in full previously (*Clarkson et al., 2017*; *Han et al., 2019*; *McQuillan et al., 2019*). In brief, ~69% of KNDy neurons express GCaMP6 with this AAV approach and 96% of all GCaMP-expressing cells in the ARN are kisspeptin neurons. Freely behaving AAV-injected mice were connected to the fiber photometry system for 4–24 hr using a fiber optic patch cord with fluorescence signals measured using a scheduled mode (5 s on, 15 s off). SEs were defined as abrupt peaks in fluorescence greater than 10% of maximum signal. Processed fluorescence signals were calculated as $\Delta F/F$ (%) = 100 $\times$ (Fluorescence − basal Fluorescence)/Fluorescence. To examine the relationship between SEs and pulsatile LH secretion, ~1.5 hr of photometry recording was coupled with 3-5 min interval tail-tip blood sampling (4 μL) as reported previously (*Clarkson et al., 2017*; *Han et al., 2019*; *McQuillan et al., 2019*). Levels of LH were measured by ELISA (*Steyn et al., 2013*), with an assay sensitivity of 0.04 ng/mL and intra-assay coefficient of variation of 9.3%.

## Acknowledgements

Studies were supported by the New Zealand Health Research Council and Wellcome Trust. Dr. A Caraty and Dr. P Ciofi (University of Bordeaux) are thanked for generous gifts of antisera. The authors thank Dr. Karl Iremonger for commenting on an earlier version of the manuscript.

## Additional information

### Funding

| Funder | Author |
| --- | --- |
| Health Research Council of New Zealand | Allan Edward Herbison |
| Wellcome Trust | Allan Edward Herbison |

The funders had no role in study design, data collection and interpretation, or the decision to submit the work for publication.

### Author contributions

Xinhuai Liu, Formal analysis, Investigation, Methodology, Writing - original draft; Shel-Hwa Yeo, Formal analysis, Investigation, Methodology, Writing - original draft, Writing - review and editing; H James McQuillan, Sabine Hessler, Formal analysis, Investigation, Writing - original draft; Michel K Herde, Formal analysis, Investigation; Isaiah Cheong, Robert Porteous, Investigation; Allan E Herbison, Conceptualization, Supervision, Funding acquisition, Methodology, Writing - original draft, Project administration, Writing - review and editing

## Author ORCIDs

Michel K Herde http://orcid.org/0000-0002-2324-2083
Sabine Hessler http://orcid.org/0000-0002-4177-4825
Allan E Herbison https://orcid.org/0000-0002-9615-3022

## Ethics

Animal experimentation: All animal handling and experimental protocols were undertaken as approved by the Animal Welfare Ethics Committees of the University of Otago, New Zealand (96/2017) or the University of Cambridge, UK (P174441DE).

## Decision letter and Author response

Decision letter https://doi.org/10.7554/eLife.62455.sa1
Author response https://doi.org/10.7554/eLife.62455.sa2

## Additional files

### Supplementary files
• Transparent reporting form

### Data availability

All data generated or analysed during this study are included in the manuscript.

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
