## [Decision Letter]

**Acceptance summary:**

KNDy neurons in the arcuate hypothalamus use kisspeptin, neurokinin B, dynorphin and glutamate as neurotransmitters to synchronize each other and regulate GnRH release that is essential for fertility. The authors of this paper use contemporary genetic and viral transduction techniques to reveal that KNDy neurons use neurokinin B, dynorphin and glutamate (but not kisspeptin) to regulate themselves but only kisspeptin to regulate episodic GnRH release at their dendrons.

**Decision letter after peer review:**

Thank you for submitting your article "Highly redundant neuropeptide volume co-transmission underlying episodic activation of the GnRH neuron dendron" for consideration by *eLife*. Your article has been reviewed by three peer reviewers, including Richard D Palmiter as the Reviewing Editor and Reviewer #1, and the evaluation has been overseen by Catherine Dulac as the Senior Editor. The following individual involved in review of your submission has agreed to reveal their identity: Stephanie Padilla (Reviewer #3).

The reviewers have discussed the reviews with one another and the Reviewing Editor has drafted this decision to help you prepare a revised submission.

Summary:

The importance of kisspeptin signaling from arcuate KNDy neurons (expressing kisspeptin, neurokinin B, dynorphin and glutamate) for fertility is well established. KNDy neurons are thought to be critical for the episodic release of LH by acting on GnRH-neuron terminals in the median eminence. A question posed here is whether kisspeptin is the only transmitter signaling onto GnRH terminals (referred to here as dendrons) in the median eminence. Some evidence suggests that the KNDy neuropeptides can be packaged into individual vesicles; thus, it is possible that only those vesicles containing kisspeptin travel to the median eminence. Alternatively, it is possible that all peptides and glutamate are released in the median eminence, but only receptors for kisspeptin are present there. To address this issue, the authors express a calcium indicator in GnRH dendrons and determine which transmitters can generate a calcium signal. They show that only kisspeptin can do so and go on to demonstrate that in the absence of kisspeptin (using KO mice), no signal is generated. This is an important result but does not completely distinguish between the two hypotheses.

Essential revisions:

1) The authors show that neither neurokinin, dynorphin nor glutamate can give rise to a calcium signal in the dendrons; however, these authors previously showed (Herde et al., 2013) that puffing glutamate onto GnRH dendrons can elicit action potentials. Resolution of this discrepancy is needed.

2) While the lack of calcium response argues that the receptors for various transmitters are not functional in GnRH dendrons, that alone does not answer the question of whether kisspeptin is the only transmitter released there. An effort to examine that aspect would make this story more interesting.

The authors should address the concerns above as well as the editorial issues raised by the reviewers including giving credit to studies from other labs that bear on this topic.

Reviewer #1:

The authors of this high-quality paper use contemporary viral/genetic technologies to show that KNDy neurons in the ARN regulate GnRH release in median eminence (ME) via kisspeptin signaling only, even though they release all their transmitters there. They monitor GCaMP fluorescence in GnRH dendrons to establish that kisspeptin signals there, but NKB, Dyn and GLU do not, whereas these 3 transmitters signal onto Kiss1-neuron cell bodies, while kisspeptin does not. They also show that loss of kisspeptin signaling in ME prevents LH release.

1) Figure 6A Authors should compare dF/F trace of Kiss1^Cre -/-^ with +/- mice, rather than referring to unpublished results.

2) The authors say, "As such, it is interesting to consider whether the episodic release of NKB and dynorphin from KNDy varicosities in the region of the ventrolateral ARN may impact on other ARN neuronal cell types." It is equally interesting to consider the possibility that KNDy neurons release all their neurotransmitters in the ME and NKB, Dyn and Glu may signal to non-GnRH neurons. It would be useful to include references documenting that NKB, Dyn and GLU are released in ME, even if kisspeptin is the only molecule that can signal to GnRH dendrons. If references do not exist, would it be possible to express GCcMP6 non-specifically ME and express ChR2 in Kiss1^Cre-/-^ KNDy neurons to show that cells in ME can respond to the other transmitters released by KNDy-neuron activation. Antagonists could then be used to establish which transmitters are released there.

3) It is also interesting to consider how Kiss1 receptors are trafficked to the dendron but neither neurokinin B receptors nor kappa opioid receptors. Some discussion of this cell trafficking problem is warranted.

Reviewer #2:

In this manuscript Liu and co-workers use in vitro and in vivo experiments to explore KNDy neuronal input onto GnRH nerve-fibers (called dendrons) in the arcuate nucleus median eminence area. The main strength of this work is the in vivo photometry experiment to activate ARN Kiss1 neurons combined with tail blood sampling for measurements of plasma LH as substitute for GnRH secretion. It is well known that Kiss1 deletion causes infertility. In addition, it is known that in some Kiss1^Cre^ mouse models homozygous animals are designed to be infertile, including the mouse model used in the current study.

1) Using the infertile homozygous Kiss1^Cre^ mouse, the authors showed that the lack of kisspeptin eliminates LH pulses following photometry stimulation in vivo of KNDy neurons, indicating that kisspeptin is responsible for LH pulses and is the main output signal from KNDy neurons onto GnRH terminals in the ME area. They also used this animal model to show that the absence of kisspeptin did not affect the synchronous firing of KNDy neurons, illustrating that kisspeptin is not involved in synchronous firing and that synchronous firing alone does not maintain fertility. However, previous studies both in vivo (Wakabayashi et al., 2010) and in vitro (Navarro et al., 2009, Qiu et al., 2016) had already provided substantial evidence for kisspeptin being the main output signal onto GnRH neurons and that NKB and dynorphin are responsible for synchronous firing.

2) It is interesting that although KNDy neurons release the peptides kisspeptin, NKB and dynorphin as well as the classical neurotransmitter glutamate, only kisspeptin was able to activate GnRH dendrons in the ME area. This is surprising since this group has shown previously (Herde et al., 2013) that both GABA and glutamate can depolarize GnRH distal dendrons. Specifically, they showed that puff application of glutamate (500 µM) on distal dendrons in vitro elicited bursts of action potentials. Currently, the authors used a similar concentration of glutamate applied in vitro and found no effect on Dendron calcium activity. Clearly further experiments are needed to sort out these differences.

Overall, although this manuscript report some compelling in vivo studies to ascertain the specific role of kisspeptin in the GnRH distal Dendron and confirm the role of NKB and dynorphin on synchronous firing, it is of limited scope and new information and therefore falls short of what one would expect for a Research Article in *eLife*.

Reviewer #3:

Authors aim to test the presence and functional significance of KNDy co-transmission at the GnRH distal dendrites in the ventrolateral ARN. The authors use expansion microscopy to score synaptic connections between KNDy and GnRH distal dendrites. Next, they use ex-vivo slice imaging to report the Ca^2+^ transients of GnRH distal dendrons during pipette application of candidate neurotransmitters. The authors go on to investigate the functional role of kisspeptin on the pulsatile firing of KNDy neurons and the subsequent release of LH using a combination of fiber photometry and repeated blood sampling. This manuscript is a continuation of a large body of work from this laboratory. Most of the techniques used here have been previously published by this group and are at the cutting edge of this research field. As a reviewer I have two points for the authors to consider in revision:

1) In 2016 Qi, Nestor et al. evaluated the mechanistic properties of synchronous firing of KNDy neurons. Along with this, they demonstrated that the influence of NKB on GnRH neurons was indirect and mediated by kisspeptin from KNDy neurons. Given this, I think it is important for the authors to more specifically compare and contrast the work from Qui, Nestor et al., 2016. While the authors do cite the manuscript, the findings are not thoroughly compared.

2) The authors show that NKB was sufficient to induce [Ca^2+^] in KNDy neurons, but not in GnRH dendrons. Given this, I found it curious that a delayed, indirect, spike was not observed in (Figure 2 A,B) from KNDy induction. Can the authors clarify this?

---

## [Author Response]

Essential revisions:1) The authors show that neither neurokinin, dynorphin nor glutamate can give rise to a calcium signal in the dendrons; however, these authors previously showed (Herde et al., 2013) that puffing glutamate onto GnRH dendrons can elicit action potentials. Resolution of this discrepancy is needed.

The studies reported in the Herde et al., paper represent the early days of our dendron work when it was only possible to explore the dendron in the rare GnRH neurons (<1%) that have their cell bodies located close to the median eminence. With the move to confocal GCaMP6 imaging, as reported here, we can now investigate all dendrons regardless of the location of their cell body. The Herde studies reported that puffs of glutamate around the median eminence evoked membrane depolarization and one or two spikes in GnRH neuron cell bodies close to the median eminence. However, we noted and reported (Figure 6B,F of that paper) a marked gradient in response with glutamate efficacy becoming markedly less as we puffed further and further away from the cell body; in essence, any glutamate puff greater than 350 microns from the cell body failed to evoke a response. Our present observations come from dendrons that will be several 1000s of microns from their cell body. Hence, it seems very likely that there is falling gradient of glutamate receptor expression along the dendron and that, while glutamate will activate those rare dendrons coming from GnRH neuron somata close by the median eminence, 99% of dendrons around the median eminence have no glutamate receptors. We have now noted this in the Discussion.

2) While the lack of calcium response argues that the receptors for various transmitters are not functional in GnRH dendrons, that alone does not answer the question of whether kisspeptin is the only transmitter released there. An effort to examine that aspect would make this story more interesting.

We have undertaken a variation of the experiment suggested by reviewer 1 in which the activity of un-identified neurons in the vicinity of the GnRH neuron dendrons in the ventral arcuate nucleus were recorded while activating KNDY projections with ChR2. We find that just over 25% of these neurons respond with either excitation or inhibition to 10-20 Hz KNDy activation and that the majority of responses could be blocked by NK3R or kappa opioid receptor antagonists. This provides key evidence supporting the hypothesis that kisspeptin, NKB and dynorphin are all released from KNDy nerve terminals in the ventral arcuate/median eminence region but that selective receptor expression results in only kisspeptin being active at the GnRH neuron dendron.

The authors should address the concerns above as well as the editorial issues raised by the reviewers including giving credit to studies from other labs that bear on this topic.

The editorial issues have all been addressed. We had cited the Qui et al. paper from the Kelly laboratory five times in the original manuscript. We now add further commentary in the Discussion stating how that paper contributes to the present conclusions.

Reviewer #1:The authors of this high-quality paper use contemporary viral/genetic technologies to show that KNDy neurons in the ARN regulate GnRH release in median eminence (ME) via kisspeptin signaling only, even though they release all their transmitters there. They monitor GCaMP fluorescence in GnRH dendrons to establish that kisspeptin signals there, but NKB, Dyn and GLU do not, whereas these 3 transmitters signal onto Kiss1-neuron cell bodies, while kisspeptin does not. They also show that loss of kisspeptin signaling in ME prevents LH release.1) Figure 6A Authors should compare dF/F trace of Kiss1^Cre -/-^ with +/- mice, rather than referring to unpublished results.

This information is given in parts B and C of this figure where two examples of GCaMP6 photometry are provided for intact Cre-/- and OVX Cre+/- mice.

2) The authors say, "As such, it is interesting to consider whether the episodic release of NKB and dynorphin from KNDy varicosities in the region of the ventrolateral ARN may impact on other ARN neuronal cell types." It is equally interesting to consider the possibility that KNDy neurons release all their neurotransmitters in the ME and NKB, Dyn and Glu may signal to non-GnRH neurons. It would be useful to include references documenting that NKB, Dyn and GLU are released in ME, even if kisspeptin is the only molecule that can signal to GnRH dendrons. If references do not exist, would it be possible to express GCcMP6 non-specifically ME and express ChR2 in Kiss1^Cre-/-^ KNDy neurons to show that cells in ME can respond to the other transmitters released by KNDy-neuron activation. Antagonists could then be used to establish which transmitters are released there.

See above.

3) It is also interesting to consider how Kiss1 receptors are trafficked to the dendron but neither neurokinin B receptors nor kappa opioid receptors. Some discussion of this cell trafficking problem is warranted.

While the trafficking of receptors to specific sub-compartments of a neuron is well established, we note that there is rather little firm evidence for GnRH neurons to express NKB or kappa opioid receptors at any location.

Reviewer #2:In this manuscript Liu and co-workers use in vitro and in vivo experiments to explore KNDy neuronal input onto GnRH nerve-fibers (called dendrons) in the arcuate nucleus median eminence area. The main strength of this work is the in vivo photometry experiment to activate ARN Kiss1 neurons combined with tail blood sampling for measurements of plasma LH as substitute for GnRH secretion. It is well known that Kiss1 deletion causes infertility. In addition, it is known that in some Kiss1^Cre^ mouse models homozygous animals are designed to be infertile, including the mouse model used in the current study.1) Using the infertile homozygous Kiss1^Cre^ mouse, the authors showed that the lack of kisspeptin eliminates LH pulses following photometry stimulation in vivo of KNDy neurons, indicating that kisspeptin is responsible for LH pulses and is the main output signal from KNDy neurons onto GnRH terminals in the ME area. They also used this animal model to show that the absence of kisspeptin did not affect the synchronous firing of KNDy neurons, illustrating that kisspeptin is not involved in synchronous firing and that synchronous firing alone does not maintain fertility. However, previous studies both in vivo (Wakabayashi et al., 2010) and in vitro (Navarro et al., 2009, Qiu et al., 2016) had already provided substantial evidence for kisspeptin being the main output signal onto GnRH neurons and that NKB and dynorphin are responsible for synchronous firing.

Please note that photometry is used to measure activity and is not a stimulatory device. The reviewer is correct in that there is very substantial, perhaps even indisputable, evidence that kisspeptin is the main output signal to the GnRH neurons. The present paper was not addressing this issue. Rather, the ability of the co-released neurotransmitters to regulate the activity of the dendron. One current hypothesis is that co-released dynorphin and NKB shape the GnRH neuron response to kisspeptin. We find no evidence for this. We disagree with the view that there is substantial evidence for NKB and dynorphin being responsible for the synchronous firing of the KNDy neurons. Evidence shows that KNDy neurons express functional tachykinin and kappa opioid receptors (de Croft et al.) and that KNDy neurons can signal to one another using these peptides (Qui et al.), however the key question as to whether NKB and dynorphin signalling amongst KNDy neurons generates their synchronous activity remains unanswered. Studies in sheep (Goodman et al.) and goats (Wakabayashi et al) infusing antagonists into the ventricle or even whole brain regions do not address this question with sufficient selectivity or precision. We do not pretend to answer this question in this paper but rather highlight the remarkably different roles of the KNDy peptides at the KNDy cells themselves and GnRH neuron dendrons.

Reviewer #3:Authors aim to test the presence and functional significance of KNDy co-transmission at the GnRH distal dendrites in the ventrolateral ARN. The authors use expansion microscopy to score synaptic connections between KNDy and GnRH distal dendrites. Next, they use ex-vivo slice imaging to report the Ca^2+^ transients of GnRH distal dendrons during pipette application of candidate neurotransmitters. The authors go on to investigate the functional role of kisspeptin on the pulsatile firing of KNDy neurons and the subsequent release of LH using a combination of fiber photometry and repeated blood sampling. This manuscript is a continuation of a large body of work from this laboratory. Most of the techniques used here have been previously published by this group and are at the cutting edge of this research field. As a reviewer I have two points for the authors to consider in revision:1) In 2016 Qi, Nestor et al. evaluated the mechanistic properties of synchronous firing of KNDy neurons. Along with this, they demonstrated that the influence of NKB on GnRH neurons was indirect and mediated by kisspeptin from KNDy neurons. Given this, I think it is important for the authors to more specifically compare and contrast the work from Qui, Nestor et al., 2016. While the authors do cite the manuscript, the findings are not thoroughly compared.

See above.

2) The authors show that NKB was sufficient to induce [Ca^2+^] in KNDy neurons, but not in GnRH dendrons. Given this, I found it curious that a delayed, indirect, spike was not observed in (Figure 2 A,B) from KNDy induction. Can the authors clarify this?

When recording from GnRH neuron dendrons, NKB is provided as a very local and brief puff to the dendrons that would be unlikely to reach the distant KNDy cell bodies.